# Positron Emission Tomography Probes for Imaging Cytotoxic Immune Cells

**DOI:** 10.3390/pharmaceutics14102040

**Published:** 2022-09-24

**Authors:** Ala Amgheib, Ruisi Fu, Eric O. Aboagye

**Affiliations:** Comprehensive Cancer Imaging Centre, Department of Surgery and Cancer, Faculty of Medicine, Imperial College London, Hammersmith Hospital, Du Cane Road, London W12 0NN, UK

**Keywords:** immunotherapies, immune responses, PET imaging, molecular imaging, PD-1/PD-L1 targeting radiotracers, LAG-3 targeting radiotracers, OX40 targeting radiotracers, CD8 targeting radiotracers, granzyme B targeting radiotracers, IL-2 radiotracers

## Abstract

Non-invasive positron emission tomography (PET) imaging of immune cells is a powerful approach for monitoring the dynamics of immune cells in response to immunotherapy. Despite the clinical success of many immunotherapeutic agents, their clinical efficacy is limited to a subgroup of patients. Conventional imaging, as well as analysis of tissue biopsies and blood samples do not reflect the complex interaction between tumour and immune cells. Consequently, PET probes are being developed to capture the dynamics of such interactions, which may improve patient stratification and treatment evaluation. The clinical efficacy of cancer immunotherapy relies on both the infiltration and function of cytotoxic immune cells at the tumour site. Thus, various immune biomarkers have been investigated as potential targets for PET imaging of immune response. Herein, we provide an overview of the most recent developments in PET imaging of immune response, including the radiosynthesis approaches employed in their development.

## 1. Introduction

Over the years, various immunotherapy approaches have emerged as powerful treatment options for cancer. These approaches aim to induce an anti-tumour immune response by redirecting or stimulating the patient’s immune system to attack cancer cells. The most common type of cancer immunotherapy is passive immunotherapy, which involves administration effector molecules, including monoclonal antibodies, or the adoptive transfer of lymphocyte-activated killer cells or cytotoxic T lymphocytes that acts to enhance existing anti-tumour immune responses. Another type of cancer immunotherapy is active immunotherapy, which involves administration of agents, such as interferons, interleukins (e.g., IL-2, IL15, and IL-12), vaccines, and genetically engineered T cells, to direct the immune system into taking an active role in attacking the cancer cells.

The clinical success of immune checkpoint monoclonal antibodies led to the approval of several therapeutic agents by the Food and Drug Administration (FDA) and European Medicines Agency (EMA) [1,2]. Cellular therapies, such as chimeric antigen receptor (CAR) T cells, have demonstrated promising clinical responses which have accelerated their approval for B cell lymphoma and B cell acute lymphoblastic leukaemia [3,4,5,6]. Despite the promising outcomes of these immunotherapeutic approaches, their clinical efficacy remains limited to a subgroup of patients, and many patients experience side effects. This can be attributed to complex interactions in the tumour microenvironment (TME), physical barriers that prevent infiltration of immune cells, upregulation of inhibitory pathways, and tumour heterogeneity and adaptability. Therefore, visualizing and monitoring immune responses may improve patient stratification and enable the identification of non-responders during the course of therapy.

Assessment of immune responses by measuring circulating levels of cytokines, lymphocytes, and immunoglobulins in blood samples or biopsies of tumour tissues is invasive and provides insufficient data on the status of infiltrating immune cells [7]. In comparison, molecular imaging approaches enable non-invasive visualisation and monitoring of immune responses. Methods, such as direct cell labelling of T cells by fluorescent agents, magnetic resonance imaging (MRI) contrast agents, bioluminescent, or radiolabelled probes, are associated with some biological alterations, which may limit their clinical translation. These limitations include the dilution of imaging agents upon cell death, potential toxicity to therapeutic cells, and restricted longitudinal imaging [8,9,10,11,12]. In contrast, T cell-targeted probes prepared by radiolabelling small molecules hold great clinical translation potential.

Positron emission tomography (PET) imaging is a powerful imaging technique that possess high sensitivity, quantitative capability, and limitless depth of tissue penetration. Thus, PET imaging is suitable for tracking T cells in clinical settings. Using T cell-specific probes, PET imaging can visualize the homing and accumulation of T cells at the tumour site. To date, few PET radiotracers have been developed for imaging cytotoxic T cells. In this review, we discuss the recent developments in cancer immunotherapy, and the use of PET imaging in predicting and evaluating immune response to cancer immunotherapy. We also discuss radioisotopes and synthesis methods employed in the development of immuno-PET imaging probes.

## 2. Immuno-Oncology

Cancer immunotherapy is an attractive strategy that aims to trigger an immune response against cancer cells, inhibiting their growth. The immune system plays two critical roles; it can suppress and/or promote tumour growth, a process known as immunoediting. During early stages of tumour development, the innate and adaptive immune system eliminates transformed cells that have escaped programmed cell death or repair mechanisms. However, rare subclones of tumour cells may survive this phase and progress into the equilibrium phase, where tumour growth is limited. Persistent activation of the immune system, along with the genetic instability of tumour cells, leads to the selection of tumour subclones with reduced immunogenicity that are capable of evading immune recognition and elimination. These tumour subclones exhibit modifications, such as loss of antigen presentation or increased expression of inhibitory immune checkpoint molecules. Various immunotherapies have been developed to relieve the immunosuppressive tumour microenvironment and subsequently trigger an anti-tumour immune response.

Cancer immunotherapy is divided into the following two categories: active and passive immunotherapy. Active immunotherapy aims at the induction of an endogenous, long-lasting tumour antigen-specific immune response. In addition, the anti-tumour response can be further enhanced via non-specific stimulation of the immune system using cytokines. Another strategy to induce in vivo stimulation is through vaccination with tumour antigens [13,14]. Passive cancer immunotherapy provides a tumour antigen-specific immune response by supplying high amounts of effector molecules, such as tumour-specific antibodies. However, passive cancer immunotherapy is short-lived and, thus, requires repeat applications. Over the years, the field of cancer immunotherapy has witnessed notable breakthroughs that have improved patients’ overall survival. Monoclonal antibodies, cytokines, immune checkpoint inhibitors (ICIs), and adoptive cellular therapy (ACT), such as chimeric antigen receptor (CAR) T cell therapy, are promising methods for treating cancer.

### 2.1. Immune Checkpoint Inhibitors

Immune checkpoint molecules play a critical role in regulating immune cell activation. Following antigen recognition by a T cell receptor (TCR), a secondary co-stimulatory signal is necessary to trigger T cell activation. This is mediated via the ligation of co-stimulatory molecules, such as CD28, expressed on T cells. Upon T cell activation, co-inhibitory molecules, such as CTLA-4 and PD-1, are recruited to the immunologic synapse to halt T cell activation and maintain physiologic immune responses (Figure 1). To avoid immune destruction, cancer cells may express more inhibitory antigens. ICIs can, therefore, block the inhibitory checkpoints, which in the presence of a co-stimulatory signal, results in the stimulation of effector cells. ICIs do not result in the direct killing of cancer cells, but they support the host’s immune system in re-enhancing anti-tumour immune response. The first ICI to be approved by the FDA was ipilimumab (anti-CTLA-4 antibody) for treating melanoma patients. Subsequently, three PD-1 and PD-L1 inhibitors were approved; a large number have since been approved [2,15,16,17,18,19].

Despite the clinical success of ICI therapy, a large number of cancer patients show no response or resistance to ICI therapy [20,21,22]. This has led to continuous efforts in identifying and evaluating other potential immune checkpoint targets. Beyond CTLA-4 and PD-L1/PD1, one target that is actively being studied is the lymphocyte activation gene-3 (LAG-3), which has been associated with the exhaustion of tumour-infiltrating T cells, as a mechanism of resistance to certain immunotherapies. A number of reports have expounded that the combinatorial blockade of LAG-3 and PD-1 pathways synergistically enhance anti-tumour immunity in some solid tumours [23,24]. This led to the recent FDA approval of Opdualag, a combination treatment of relatlimab (anti-LAG3 antibody) and nivolumab (anti-CTLA-4 antibody), for patients with inoperable or metastatic melanoma. Opdualag was found to prolong progression-free survival (10.1 months) compared to those receiving nivolumab monotherapy (4.6 months) [25]. Although nivolumab and ipilimumab combination therapy has yielded similar clinical results, Opdualag was found to be associated with fewer side effects. The effect of Opdualag is currently being evaluated in clinical trials of other cancers, such as lung, liver, and colorectal cancer [26,27,28].

The T cell immunoreceptor with immunoglobulin and immunoreceptor tyrosine-based inhibitory motif domains (TIGIT) is an immune receptor that is mostly expressed on T cells and natural killer (NK) cells. Varying levels of TIGIT are detected on different subsets of T cells. Furthermore, TIGIT is an inhibitory immune checkpoint molecule that plays role in modulating tumour-targeted T cell response and, thus, serves as potential target for immune checkpoint inhibition. The main ligand for TIGIT is CD155, which is upregulated on cancer cells and is also expressed on tumour-infiltrating myeloid cells. In pre-clinical studies, anti-TIGIT mAb monotherapy was found to inhibit tumour growth and prevent metastasis. In myeloma mouse models, blocking of TIGIT was found to reduce tumour burden and prolong overall survival [29]. Furthermore, a dual blockade of TIGIT and PD-1/PD-L1 pathways was shown to result in tumour rejection even in tumour models resistant to anti-PD-1 therapy. Blockade of TIGIT not only enhances the response of effector T cells but also NK cell responses, and it reduces the suppressive capacity of regulatory T cells (Tregs). At present, multiple clinical studies are investigating the safety and therapeutic value of TIGIT blockade in combination with other ICIs [30,31,32].

### 2.2. CAR-T Cell Therapy

Another promising approach in the field of cancer immunotherapy is CAR-T cell therapy, whereby T cells are genetically modified to express synthetic receptors that allow them to recognise tumour-associated antigens (TAAs) presented by human leukocyte antigen (HLA; the major histocompatibility complex [MHC] in humans) molecules. CAR-T cells consist of an extracellular antigen-binding domain that is linked to an intracellular signalling domain CD3ζ (first generation CARs). In second- and third-generation CARs, CAR-T cell activity is augmented by the addition of one or more co-stimulatory domains. Fourth-generation CARs possess a constitutive or inducible expression of soluble cytokines or co-receptors [33,34]. To date, six CAR-T cell therapies have been approved by the FDA for treating lymphomas, multiple myeloma, and some types of leukaemia [3,35,36,37,38,39].

Although CAR-T cell therapy has exhibited promising efficacy, not all patients achieve complete responses. In fact, durable remission is not guaranteed, and 30–60% of patients are reported to relapse following treatment with CD19 CAR-T cells, which may be due to antigen escape [40]. Additionally, continuous antigen exposure results in T cell exhaustion, which is associated with poor responses in patients receiving CAR-T cell therapy. Furthermore, CAR-T cell therapy is less effective in solid tumours due to poor trafficking and infiltration into tumour tissue. The immunosuppressive microenvironment, characterised by the presence of immune suppressor cells and immunosuppressive cytokines, presents another hurdle for treating solid tumours. One proposed strategy to improve the therapeutic efficacy of CAR-T cells is combination therapy.

The ZUMA-6 trial (phase 1) demonstrated that combining anti-CD19 CAR-T cell therapy with PD-L1 therapy is safe and promising in treating patients with refractory aggressive non-Hodgkin lymphoma [41]. In a small study of four children with refractory acute lymphoblastic leukaemia, treatment with PD-1 antibodies was found to restore CAR-T cell function [42]. An alternative strategy that has been evaluated in pre-clinical studies involves the use of CAR-T cells engineered to release an immune checkpoint blockade single-chain variable fragment (scFv), which was found to enhance the survival of PD-L1 positive tumour-bearing mice [43]. In addition, this approach of localised delivery of ICIs may decrease the adverse effects associated with systemic ICIs. Nevertheless, the clinical efficacy and safety of this approach are yet to be determined.

### 2.3. γδ T Cell Therapy

Another adoptive cell therapy that has gained considerable attention as an attractive candidate for cancer immunotherapy is the use of γδ T cells, which are a minor population of peripheral lymphocytes [44]. Indeed, γδ T cells have been demonstrated to exhibit potent anti-tumour effects, and possess unique characteristics, such as the ability to recognise antigens independently of the HLA class I molecules [45]. Unlike conventional αβ T cells, γδ T cells are activated by phosphoantigens (PAgs), such as isopentenyl pyrophosphate (IPP), which is an intermediate product in the mevalonate pathway [46]. It is now recognised that IPP binds to the intracellular domain of BTN3A1, expressed in target cells, inducing a conformational change in its extracellular domain that is recognised by γδ TCR [47,48]. Recently, another member of BTN family, BTN2A1, has also been reported to be as essential for the activation γδ T cells; BTN2A1 is proposed to interact with the germline-encoded regions of the Vγ9 chain of the TCR, acting together with BTN3A1 to trigger γδ T cell response to PAgs [49,50]. Furthermore, γδ T cells co-express receptors of the innate immune cells, such as by activating NK receptors and certain Toll-like receptors. They also share many functions with the conventional αβ T cells, such as their ability to produce cytokines and interleukins. Pre-clinical studies demonstrated γδ T cell-based therapy to possess a potent anti-tumour effect [51,52,53]. Nevertheless, their clinical efficacy was found to be inconsistent in multiple phase 1 clinical trials despite their safety; this can be attributed to multiple factors, such as resistance of tumour cells to γδ T cell-cytotoxicity, and/or the poor activation and infiltration of γδ T cells [54,55,56,57]. Currently, multiple phase 1/2 clinical trials are evaluating the safety and efficacy of ex vivo expanded allogenic γδ T cells. Additionally, to improve the efficacy of γδ T cells, Oberge et al. demonstrated the use of bispecific antibodies in triggering γδ T cells cytotoxicity against HER2-expressing cancer cells in vitro and in vivo [58]. Another strategy proposed to overcome the therapeutic limitations of γδ T cells is the development of CAR-γδ T cells [59,60].

### 2.4. Cytokines

Cytokines, which are small proteins secreted by different immune cells, play a significant role in cancer immune cycle. This includes antigen presentation on cancer cells, priming and activation of T cells, infiltration of effector cells into tumour tissue, and cancer cell death [61]. Furthermore, cytokines are involved in mediating immune cell differentiation, which determines the effect of anti-cancer immunity. The most studied cytokines in cancer therapy are granulocyte-macrophage colony-stimulating factor (GM-CSF), vascular endothelial growth factor (VEGF), interluekin-2 (IL-2), granulocyte colony-stimulating factor (G-CSF), and interferon gamma (IFN-γ). Currently, Aldesleukin (recombinant human IL-2) is approved by the FDA for treating patients with metastatic melanoma and metastatic renal cell carcinoma [62]. Both CSF and IL-2 promote the proliferation and differentiation of immune cells, and consequently can be used to enhance anti-cancer immunity. Furthermore, IFN-γ can be used to directly inhibit the proliferation of cancer cells and enhance anti-tumour immunity [62]. Based on the outcomes of multiple clinical trials, the efficacy of many cytokines as anti-cancer therapeutic agents is limited, which may be associated with their short circulation time [61,63]. Thus, frequent administration is required to achieve long-lasting therapeutic effect; however, this is very likely to result in adverse events [64]. To overcome these limitations, many studies investigated the use of nanomaterials as potential cytokine carriers due to their preferential accumulation at the tumour site, aqueous solubility, and prolonged circulation time. Polyethylene glycol (PEG) molecules were also demonstrated to be effective carriers for cytokine delivery. One example is Bempegaldesleukin (NKTR-214), which is an engineered IL-2 receptor (IL-2R) agonist with an average of six releasable PEG molecules [65]. In patients with metastatic melanoma and renal cell cancer, Bempegaldesleukin monotherapy was found to expand peripheral and intra-tumoural infiltration of cytotoxic T cells without affecting the population of Tregs and without causing serious toxicity [66]. However, phase 3 clinical trial with Bempegaldesleukin in combination with nivolumab reported no additional clinical benefit and, as a result, patient enrolment was discontinued [67]. Another engineered IL-2 that showed promising preclinical data is THOR-707, which has a PEG molecule attached to block its binding to the CD25 subunit of the IL-2R. The safety and therapeutic activity of THOR-707 is currently being studied in a phase 1/2 clinical trial in patients with advanced or metastatic solid tumours [68].

#### 2.4.1. Challenges Associated with Cancer Immunotherapy

The promising clinical benefits observed in some patients treated with cancer immunotherapy indicate its feasibility in restoring effective anti-tumour immune surveillance. Cancer vaccines, another type of immunotherapy, have a preventive and therapeutic potential and, thus, may provide long-term immunity against cancer recurrence. To date, three vaccines are approved for treating metastatic melanoma, early-stage bladder cancer, and metastatic castration-resistant prostate cancer [69,70,71]. Currently, multiple phase 2 clinical trials are evaluating the efficacy of these vaccines for other types of cancers. Nevertheless, the clinical effect of cancer vaccines and other immunotherapies is only observed in small group of patients. This may be due to tumour heterogeneity, treatment history, the underlying immunosuppressive biology of cancer, and variability in cancer type and stage [21,72]. Identification of predictive or prognostic biomarkers may enable the selection of patients that will most likely benefit from cancer immunotherapy.

To date, there are three FDA-approved positive predictive biomarkers, namely PD-L1 expression, tumour mutational burden (TMB), and microsatellite instability (MSI) [72]. The first, PD-L1, is one of the biomarkers that is robustly investigated in predicting response to ICIs. Data suggest that tumours with high PD-L1 expression are associated with better response and survival rates with PD-1/PD-L1 ICIs therapy. However, two clinical trials studying nivolumab in metastatic melanoma patients showed that 20–30% of PD-L1 negative patients responded to therapy [73,74]. Thus, the discovery of other biomarkers may improve treatment prognosis.

Another biomarker, MSI, develops as a result of defects in the DNA mismatch repair pathway that leads to the accumulation of many mutations within microsatellite regions. Tumours with high MSI exhibit increased mutational burden, resulting in the infiltration of T cells in the TME. Furthermore, TMB and the presence of inflamed gene signatures have been reported to positively correlate with response to PD-1/PD-L1 blockade. Patients with high TMB were found to have better overall survival when treated with PD-1/PD-L1 ICIs. In addition, higher TMB associates with a greater probability of displaying neoantigens on HLA molecules on the surface of cancer cells, triggering cytotoxic T cell-dependent immune response [75]. Therefore, analysis of TMB, and the presence of TILs along with expression of PD-1/PD-L1, may enable the identification and selection of responders to ICI therapy [76]. In addition, cancer immunotherapies are very expensive to develop and administer, which limits their use in specific patient populations. Thus, more cost-effective techniques need to be developed and applied to enable the accessibility of cancer immunotherapies to a broader range of patients. Consequently, the identification of novel clinical and molecular predictive biomarkers may enable the selection of patients that are likely to benefit from expensive immunotherapy treatment.

The efficacy of cancer therapies is assessed by measuring the tumour volume, analysis of tissue biopsies, and performing peripheral blood assays. However, changes in tumour volume may prove misleading, since the influx of effector immune cells into TME often contributes to increased volume, which is a phenomenon known as pseudoprogression [77,78]. In addition, tissue biopsies taken post-treatment is an invasive method that is dependent on the accessibility of the tumour and often fails to account for tumour heterogeneity. Furthermore, tissue biopsies may not reflect the complex interactions between tumour and immune cells. Immunohistochemistry (IHC) is a technique that is routinely performed to stain for immunoregulatory proteins in clinical tissue biopsies. However, IHC limits accurate classification of both cell type and function, since staining for more than two markers requires a careful selection of primary antibodies or the use of consecutive tissue sections, which is sometimes difficult to obtain due to low tissue availability in some samples. Newer technologies are emerging to address some of these challenges, such as multiplex IHC (mIHC) [79,80]. Nevertheless, clinically accessible mIHC only enables staining for a limited number of markers. Other techniques, such as peripheral blood assays, which are commonly used to reveal the diversity of immune infiltrates in TME, do not reflect the dynamics and spatial information that are required to monitor immune responses to treatment. Therefore, there is a demand for developing diagnostic and predictive methods to detect and monitor anti-tumour immunity.

Non-invasive molecular imaging approaches that enable monitoring of systemic and intra-tumoural alterations in immune cell localisation may increase our understanding of the dynamics of various immunotherapeutic strategies. There are multiple techniques for non-invasive cell tracking, such as ex vivo cell labelling and radiolabelled metabolic probes. However, these strategies are associated with potential toxicity to the therapeutic cells, dilution of imaging agents upon cell death, and restricted longitudinal imaging that may limit their clinical translation. In contrast, T cell specific probes made by labelling antibodies or small molecules possess great translation potential. The imaging technologies employed include magnetic resonance imaging (MRI), computer tomography (CT), positron emission tomography (PET), and single photon emission computed tomography (SPECT). Unlike optical cell-tracking methods, PET imaging has high sensitivity and spatial resolution and, thus, may provide insight into immune activity in TME, and subsequently present a tool for evaluating treatment strategies.

#### 2.4.2. PET Imaging of Immune Cells

PET imaging method is a widely used non-invasive clinical diagnostic technique. It tracks the spatial distribution of the PET radiotracer by detecting the 511 keV gamma rays from the positron/electron annihilation events following positron decay of the radionuclide. Owing to the highly distinctive signal, very low levels of radioactivity can be detected. This makes PET imaging a highly sensitive imaging technique [81]. Moreover, the distribution of the PET tracer is related to a specific biological process [82]. Therefore, it does not only provide the spatial information of the tracer, but it can also quantitatively represent the relevant biological function. Using this technique, a whole-body visualization of the immune response could be generated with detailed functional and dynamic information without invasive biopsies.

While the PET imaging scanner technology has made huge leaps during the last few years [83], there is a critical role for developing the optimal PET radiotracer, and radiolabelling plays a crucial part in the development of a PET tracer. Depending on the physical properties, such as half-life, decay characteristics, and labelling chemistry, different methods are employed to enable radiolabelling [84]. Commonly used PET radionuclides are summarized in Table 1. Most of the compounds used in PET imaging of immune response are radiolabelled with ^18^F, ^68^Ga, ^89^Zr, and ^64^Cu. Some PET radionuclides have a short half-life (such as ^11^C t_1/2_ = 20.4 min) and, therefore, are the reserve of specialist imaging facilities with on-site cyclotrons. Additionally, many other PET radionuclides cannot be easily sourced commercially, such as ^44^Sc and ^124^I (Table 1). In general, the radionuclides can be grouped into two major categories, namely non-metal and metal PET radionuclides.

A non-metal radionuclide, namely ^18^F, is the most widely used PET radionuclide. Its main clinical application is to detect cancer through labelling of glucose-mimicking [^18^F] fluorodeoxyglucose ([^18^F]FDG) by exploiting the high metabolic rate of cancer [85]. However, for imaging immune response, many biomolecules, such as peptides, antibody fragments, and full-length antibodies are used as the targeting motif. The high temperature and high organic solvent environment are not suitable for these delicate structures. Thus, many ^18^F radiolabelled compounds depend on the ‘prosthetic’ group approach, involving small prosthetic molecules that are labelled with ^18^F prior to conjugation to biomolecules under mild conditions. The trade-off for the ‘prosthetic group approach’ is a lower yield and more complicated synthesis due to the requirement of extra synthesis time and additional purification.

Radiometals, such as ^64^Cu (t_1/2_ = 12.7 h), have long been used for PET imaging [86]. Many radiometals provide longer decay half-life compared to their non-metal counterparts, which is desirable for labelling larger biomolecules with a longer biological half-life. In addition, the recent commercial availability and improvement of the ^68^Ge/^68^Ga generators has allowed ^68^Ga (t_1/2_ = 68 min) to be used more widely for smaller molecules, such as peptides [87]. During the last decade, ^89^Zr has gained a lot of attention for PET imaging owing to its 78.4 h half-live, which matches the biological half-life of full-length IgG antibodies. In contrast to non-metal radiolabelling, radiometal labelling is easier to implement and often requires much milder condition. To achieve labelling, a chelator is usually first conjugated to the antibody/fragment/peptide. Then, at the time of radiolabelling, a solution of a radiometal is added to the chelator conjugated antibody in a suitable buffer system. Depending on the type of radionuclide and chelator, the radiolabelling is achieved either at room temperature or at an elevated temperature (Table 2).

As aforementioned, there are a variety of suitable targeting motifs, ranging from small molecules, peptides, nanobodies, antibody fragments, and full-length antibodies. Each of these have their own advantages and disadvantages (Table 3). Peptides and small molecules are often derived from the nature-identical substrate or ligand of a desired target. Thus, the discovery could be accelerated if a known structure exists. These molecules often have fast pharmacokinetics (tens of minutes) and can tolerate relatively harsh radiolabelling conditions. Radionuclides with a short half-life could pair with these molecules for fast and convenient same-day PET imaging. In contrast, considerations are different when using a protein as the targeting motif. For example, full-length antibodies are naturally occurring proteins that bind antigens with high affinity and selectivity. Over the last few decades, a matured industry has been established for the generation, selection, manufacturing, and modification of monoclonal antibodies (mAbs). This makes it a desirable scaffold for developing PET tracers from these highly versatile proteins. As a PET tracer, mAbs can potentially achieve high tumour uptake. However, the relatively slow pharmacokinetics (days to weeks) of the antibody means radionuclides with a longer half-life must be used. To minimise radiation risk and simplify the diagnostic procedure, engineered antibody fragments with faster pharmacokinetics, such as diabody and minibody fragments, have been developed to allow for faster imaging and a lower radiation (effective) dose. Nanobodies are another class of domain antibody, derived from camel and llama, that produce an antibody with only the heavy chain binding domain. The advantages of nanobodies have a lot to do with their highly compact structure. With a typical molecular weight of 15 kDa, nanobodies retain the high affinity and specificity of an antibody while having much faster pharmacokinetics (a few hours). In addition, nanobodies can tolerate more challenging conditions, such as higher temperatures. Merging the advantages of small molecules and antibodies, nanobodies have become an emerging scaffold for PET tracers after its main patent expired in the late 2010s [88]. Other potential vectors include affibodies, DARPins, and affimers [89,90,91].

### 2.5. Fluorine-18 Labelled Fluorodeoxyglucose ([^18^F]FDG)

The [^18^F]FDG radiotracer is a glucose analogue that accumulates in cells with enhanced metabolic activity (Figure 2). It is routinely used in the clinic to determine tumour stage and treatment efficacy, including response to ICIs. However, [^18^F]FDG does not only target cancer cells, but it can also be taken up by immune cells, making it difficult to distinguish between tumour-related uptake from those induced by immunotherapy [92,93,94]. This led to the proposal of imaging interpretation criteria, which includes revision of the Lugano criteria and the Lymphoma Response to Immunomodulatory therapy Criteria (LYRIC) to avoid misdiagnosis [95]. Studies focusing on validating [^18^F]FDG PET/CT in early phases of immunotherapy reported high [^18^F]FDG uptake by the tumour with or without increase in tumour volume following administration of ICIs. This is likely to indicate activation of the immune system, since immunotherapy results in pseudoprogression. In addition, the enhanced infiltration of immune cells via ICI agents influences the tumour microenvironment and may promote the glycolysis of cancer cells, resulting in an increased uptake of [^18^F]FDG [96,97]. Thus, PET imaging with [^18^F]FDG, along with CT scans, may help to distinguish pseudoprogression from other responses. For instance, in a cohort of melanoma patients receiving ICI therapy, residual metabolic activity on [^18^F]FDG PET was found to be associated with residual tumour masses [98]. Dercle et al. demonstrated that a reduction in the avidity of ^18^F-FDG in the spleen and tumour of patients with relapsed or refractory Hodgkin lymphoma after 3 months of initiating ICI therapy correlated with improved prognosis [99]. Nevertheless, the low specificity of [^18^F]FDG for immune cells makes it difficult to differentiate between subsets of infiltrating immune cells and tumour cells.

Targeting of cell-surface markers by PET tracers provides increased specificity for subsets of tumour-infiltrating cells and may allow early determination of treatment efficacy. Consequently, many T cell specific PET probes targeting many different surface markers, such as CTLA-4, PD-1, CD3, and CD8, have been developed and are intensively studied (Figure 3 and Table 4).

## 3. PET Imaging of Immune Checkpoints

Different studies have evaluated the potential clinical value of PET imaging of immune checkpoint targets in the assessment of T cell dynamics, cancer diagnosis, and patient stratification prior to initiating immunotherapy. Intact monoclonal antibodies targeting immune checkpoints have been used for PET imaging studies. One example is the radiolabelling of nivolumab with zirconium-89 (^89^Zr), which showed the feasibility of such an approach in mapping PD-1 expression in humanized murine models of lung cancer [100]. Initial clinical evaluation of [^89^Zr]Zr-DFO-nivolumab demonstrated its safety and ability to quantify PD-1 expression in patients with non-small cell lung carcinoma (NSCLC) [101]. Nevertheless, larger clinical studies are required to further validate the clinical potential of using [^89^Zr]Zr-DFO-nivolumab in predicting responses to anti-PD-1 immunotherapy.

Zirconium-89 (^89^Zr) is the most widely used radiometal for PET imaging of immune response when mAb is the targeting motif. The long half-life of 78.4 h and the ease of ^89^Zr-DFO chemistry makes it very attractive for labelling antibodies and smaller antibody fragments, such as minibodies [102]. Desferrioxamine B (DFO), as the name indicates, is originally an iron chelator derived from bacteria. Zirconium coordination preference is very similar to iron, thus, making DFO a suitable chelator for ^89^Zr. As an acyclic chelator, DFO allows chelation to be achieved at room temperature (Figure 4). In contrast, elevated temperature is required by many macrocyclic chelators to affect complexation [103]. To achieve labelling, different variants of DFO were reported to be conjugated on antibodies. The first and most commonly used method is to conjugate p-NCS-Bz-DFO via thiocyanate–lysine conjugation under slightly basic condition (pH 9.0) [104]. This type of reaction results in a stable thiourea bond linkage between the DFO and lysine side chain of the antibody. However, due to availability of multiple lysines per antibody for conjugation, the resulting DFO-antibody conjugates are mixtures of the distribution of the drug to antibody ratio (DAR). Recently, Jung et al. have reported site-specific conjugated DFO via the interchain disulphate bond of an anti-PDL-1 antibody. The resulting DFO conjugate could achieve a DAR of 2, most likely due to conjugation to a pair of interchain disulphate bridges. In addition, Christensen et al. have reported a glycan conjugated DFO antibody, DFO-6E11, with an estimated DAR of 2 [105]. Post-conjugation, the DFO conjugated antibody is incubated with ^89^Zr solution in HEPES buffer (pH 6.8–7.5) for 60 min to effect radiolabelling [106], and the radiolabelled [^89^Zr]Zr-DFO-antibody is isolated using size exclusion column chromatography at the end of reaction.

Another biomarker, PD-L1, has also been extensively studied for immune response. Indeed, PET imaging studies of PD-L1 with [^89^Zr]Zr-DFO-atezolizumab (FDA approved anti-PD-L1 antibody) demonstrated a strong correlation between PD-L1 expression and clinical outcome in NSCLC patients [107]. Another monoclonal anti-PD-L1 antibody, [^89^Zr]Zr-DFO-durvalumab, demonstrated higher tumour uptake in patients with advanced NSCLC who responded to durvalumab. However, uptake of [^89^Zr]Zr-DFO-durvalumab did not correlate to tumour PD-L1 expression as determined by IHC [108]. This indicates that PET imaging may provide a more comprehensive evaluation of biomarker expression compared to IHC-assessment of biopsy samples. Nevertheless, IHC may be used in combination with molecular imaging to detect PD-L1 expression in tumour cells and various subsets of immune cells, which is a limitation of anti-PD-L1 PET tracers. Other PET tracers that have been recently evaluated in clinical studies are [^89^Zr]Zr-DFO-REGN3504 and [^89^Zr]Zr-DFO-avelumab; however, clinical data are yet to be published [109,110].

It is worth noting that anti-PD-1 and PD-L1 PET tracers are based on full-length monoclonal antibodies (mAb), which are associated with lower tumour penetration, lower tumour-to-background ratios, and slow peripheral clearance kinetics, as opposed to small molecule compounds. Consequently, nanobodies that bind to PD-L1 have been widely studied. Due to their fast peripheral clearance, PET imaging can take place as early as one hour post-injection. In addition, nanobodies can be radiolabelled with short-lived radioisotopes, thus, lowering the amount of radiation in patients. Bridoux et al. demonstrated the stability and specificity of the [^68^Ga]Ga-NOTA-(hPD-L1) nanobody for PD-L1 imaging in vivo [111]. Another study demonstrated the potential of radiolabelling non-blocking PD-L1 nanobody, [^68^Ga]Ga-NOTA-Nb109, in mapping PD-L1 expression in xenograft tumours [112,113]. Currently, [^68^Ga]Ga-THP-APN09 PET is under clinical evaluation in patients with lung cancer, melanoma, and other solid tumour undergoing anti-PD-L1 therapy [114].

The ^68^Ga radionuclide has a short 68 min half-life, and it is used for labelling of nanobodies. Traditionally, ^68^Ga is complexed with 2,2′,2′′,2′′′-(1,4,7,10-Tetraazacyclododecane-1,4,7,10-tetrayl)tetraacetic acid (DOTA) chelator. It has been used clinically with DOTA-TOC and DOTA-TATE for imaging of SSTR2 overexpressing gastroenteropancreatic neuroendocrine tumours (GEP-NETs) [115]. However, radiolabelling ^68^Ga with DOTA requires elevated temperature of close to 100 °C, which would risk proteins being denatured [116]. In recent years, the commercial availability of more suitable chelators, such as NOTA and (Tris(hydroxypyridinone) (THP), allow ^68^Ga labelling to be performed under milder conditions (37–60 °C). Typically, ^68^Ga labelling for immune response are almost exclusively paired with these two chelators [113,117,118,119].

Additionally, [^89^Zr]Zr-DFO-ipilimumab is another mAb-based PET tracer that was developed to image CTLA-4 expression. Preliminary data from an ongoing clinical study in patients with metastatic melanoma receiving ipilimumab monotherapy reported the feasibility of [^89^Zr]Zr-DFO-ipilimumab for visualizing and quantifying ipilimumab uptake in tumours [120]. Pre-clinical studies demonstrated the potential of using [^64^Cu]Cu-DOTA-ipilimumab in visualising CTLA-4 in NSCLC xenografts [121,122].

The 12.7 half-life of ^64^Cu makes it another attractive radionuclide for PET imaging. The radiolabelling of ^64^Cu is commonly paired with a DOTA chelator and can be achieved at 37 °C, and pH 5.5, for 60 min with antibodies [123,124]. There are reports that the [^64^Cu]Cu–DOTA complex could lead to high liver, kidney, and spleen uptake. Replacing DOTA with a NOTA-like chelator can lead to a more stable complex with ^64^Cu and can circumvent this issue [125,126]. However, such chelators have still not been widely used in the field of immune response monitoring. After chelation, scavengers, such as EDTA, can be used to remove the excess ^64^Cu and the resultant [^64^Cu]Cu–DOTA-antibody could be isolated via size exclusion chromatography.

To date, [^89^Zr]Zr-DFO-ipilimumab is the only PET tracer that is undergoing clinical evaluation for imaging CTLA-4. However, LAG-3 has also been studied as a potential target for PET imaging of immune checkpoints. Furthermore, [^89^Zr]Zr-DFO-REGN3767, a fully human anti-LAG-3 mAb, was shown to detect LAG-3 expression in mouse tumours [127]. At present, the safety and diagnostic potential of [^89^Zr]Zr-DFO-REGN3767 is under evaluation in a clinical study in patients with diffuse large B cell lymphoma (DLBCL) [128]. Another inhibitory immune checkpoint molecule, TIGIT, is also an interesting target for PET imaging. Additionally, [^68^Ga]Ga-NOTA-GP12, a peptide antagonist for TIGIT, was demonstrated to be safe and able to image TIGIT expression in murine models and two patients with advanced NSCLC [117]. Nevertheless, further clinical evaluation is necessary to determine its potential value in predicting and monitoring response to ICIs.

## 4. PET Imaging of CD8^+^ and CD3^+^ T Cells

The suppressive immune TME of many tumours is characterised by exhausted T cells or the absence of infiltrating lymphocytes. Therefore, the success of immunotherapy depends not only on the expression of appropriate immunotherapy targets in a tumour, but also on the cellular composition and a range of bio-active constituents present within the TME. Due to the development of many CD8^+^ and CD3^+^ PET tracers, it is becoming increasingly feasible to image effector T cells. Larimer et al. developed [^89^Zr]Zr-DFO-IgG for imaging CD3^+^ T cells. Evaluation of this PET tracer in murine models demonstrated high uptake by tumours in response to anti-CTLA-4 treatment, which correlated with a subsequent reduction in tumour size [129]. In another study, [^89^Zr]Zr-DFO-anti-CD3, imaged the distribution of homeostatic T cells, particularly TILs, in syngeneic mice bearing bladder cancer [130]. Although CD3^+^ imaging may enable the assessment of all subsets of T cells, PET imaging of CD8^+^ allows imaging of cytotoxic T cells, which play a key role in the anti-tumour immune response. Various CD8^+^ PET tracers have been developed and studied in vivo. Furthermore, [^68^Ga]Ga-NOTA-SNA006a, a nanobody-based tracer targeting CD8, exhibited rapid and persistent uptake in tumour lesions, as well as in CD8-rich tissues in humanised mouse xenografts [118]. The combination of a nanobody with a short-lived positron emitter ^68^Ga (t_1/2_ = 68 min) may minimize organ radiation exposure and consequential side effects experienced by the patient. Nevertheless, there are no data available on the potential effect of anti-CD8 nanobody binding on the activation and function of effector T cells, which is important for clinical translation.

Additionally, [^89^Zr]Zr-DFO-IAB22M2C, a humanised minibody that is biologically inert, is reported to detect CD8^+^ T cells without affecting T cell proliferation, activation, or function in mouse models [131,132]. Furthermore, in a phase 1 clinical trial, [^89^Zr]Zr-DFO-IAB22M2C was found to accumulate in lymphoid organs and tumour lesions, correlating with infiltration of CD8^+^ T cells (Figure 5 and Figure 6) [133]. However, the reported whole-body clearance of [^89^Zr]Zr-DFO-IAB22M2C was similar to a full-size antibody. Consequently, patient radiation exposure (per MBq injected activity) is expected to be significantly higher than those of small molecules. This, in turn, may potentially limit the clinical application of this imaging probe. Currently, the diagnostic and prognostic potential of [^89^Zr]Zr-DFO-IAB22M2C in imaging CD8^+^ T cells in patients treated with immunotherapy is being evaluated in multiple clinical trials; the probe remains one of the most important commercial imaging tools in immuno-oncology [134].

## 5. PET Imaging of Immune Cell Activation

The PET probes targeting CD3^+^ and CD8^+^ T cells may capture the dynamics of T cells; however, they do not provide information on the activation and functional state of T cells. The suppressive TME of many malignancies is characterised by the absence of TILs or the presence of exhausted of T cells, which induce immunotolerance. Therefore, characterising the activation and function state of infiltrating immune cells may enable a more accurate prediction of response to cancer immunotherapy. As a result, different PET probes have been developed to image markers that are upregulated on or released by activated cytotoxic immune cells. Such markers are OX40, the IL-2 receptor (IL-2R), granzyme B, and IFN-**γ**. The PET probes developed for these markers are discussed in this review.

### 5.1. OX40

The OX40 marker is a member of TNF receptor superfamily, and its expression is restricted to antigen-specific activated T cells [135]. Alam et al. developed a [^64^Cu]Cu-DOTA-AbOX40 that enables imaging of OX40 receptor to monitor activated T cell responses in a clinically relevant in situ cancer vaccine model [136]. Then, [^89^Zr]Zr-DFO-OX40, which has a longer half-life than ^64^Cu, was developed by the same group to observe the longer kinetics of immune response to a vaccine in a murine glioblastoma model treated with CpG oligodeoxynucleotide [137]. The PET scans showed uptake of [^89^Zr]Zr-DFO-OX40 in tumour lesions, as well as distant lymph nodes, indicating that the immune response initiated by the vaccine can be analysed in the whole body. Although these studies demonstrate the feasibility of imaging OX40 to detect immune cell activation, smaller constructs that are radiolabelled with short-lived isotopes may be more beneficial, as they are associated with lower radiation exposure.

### 5.2. Interleukin-2 (IL-2) Receptor

IL-2 is a small glycoprotein (~15 kDa) that is predominantly released by T cells during an immune response. Secreted IL-2 binds to IL-2R that is expressed by activated immune cells, promoting their proliferation and differentiation. IL-2R is composed of IL-2Rα (CD25), IL-2Rβ (CD122), and IL-2Rγ (CD132). IL-2Rα possess a low affinity for IL-2 (Kd~10^−8^ M) and forms a high affinity trimeric αβγ complex (Kd~10^−11^M) in the presence of IL-2Rβ and IL-2Rγ subunits. The expression of high affinity IL-2R is upregulated on immune cells upon activation and it is also expressed on Tregs [138,139]. Therefore, high affinity IL-2R represents a potential target for imaging immunosuppressive and immunostimulatory cells in TME.

Different studies have demonstrated the clinical potential of using radiolabelled IL-2 probes for detecting activated immune cells in chronic autoimmune diseases via SPECT imaging. Gialleonardo et al. have explored PET imaging of activated T cells using [^18^F]-fluorobenzoyl-interleukin 2 ([^18^F]FB-IL-2) probe [140]. This probe uses N-succinimidyl-4-[^18^F]fluorobenzoate ([^18^F]SFB) to affect radiolabelling of the recombinant IL-2 molecule, because of the simplicity of conjugating it with the lysine side chain of a peptide/protein (Figure 7A). Gialleonardo et al. have shown that using this method, recombinant IL-2 could be labelled with good yield (c.a. RCY = 10%) and purity [140]. In addition to this, Allott et al., using click-chemistry, achieved labelling of IL-2 (Figure 7B) with E-2-(((4-[^18^F]fluorobenzylidene)amino)oxy)-N-(4-(6-methyl-1,2,4,5-tetrazin-3-yl)benzyl)acetamide ([^18^F]FBoxTz). This method allows the whole IL-2 radiolabelling process to be fully automated under GMP compatible conditions with a similar yield to the [^18^F]SFB method [141,142]. Additionally, [^18^F]FB-IL-2 was shown to specifically distinguish between unstimulated and stimulated PBMCs in various murine models [140,143,144]. Although [^18^F]FB-IL-2 was found to be safe in patients with metastatic melanoma receiving ICI therapy, its tumour uptake did not correlate with treatment outcome [145]. In addition, there was no correlation between IL-2R expression and baseline uptake of [^18^F]FB-IL-2 in four tumour tissue samples. This may be due to competitive binding with endogenous IL-2, as was suggested by the authors of the study; nevertheless, further studies are required to confirm this.

### 5.3. Granzyme B

One way to monitor the function of cytotoxic T cells in response to cancer immunotherapy is by imaging granzyme B, which is a serine protease that is secreted by both cytotoxic T cells and natural killer cells to induce cancer cell death.

The granzyme B PET agent, mNOTA-GZP, is a peptide-derived compound that was first described in 2017 by Larimer et al. [146]. The peptide sequence was further developed from a tetrapeptide substrate sequence IEPD described by Thornberry et al. in 1997 [147]. To enable this sequence for PET imaging, proline was replaced by phenylaniline in the sequence (hence, IEFD) to make it an irreversible inhibitor. Then, a poly-glycine linker and NOTA chelator were added to the sequence to enable radiolabelling. The resulting probe has binding Ki of 47 nM. Additionally, ^68^Ga and [^18^F]AlF have both been used for the labelling of mNOTA-GZP. Indeed, [^18^F]AlF is an interesting emerging method for the radiolabelling immune molecules. As summarised by Archibald and Allot, its simplicity and efficiency make it a very attractive labelling method [148]. This method has the advantage of high Al–F bond energy, allowing fast complexation of [^18^F]AlF with a suitable chelator, such as 2,2′,2′′-(1,4,7-triazacyclono-nane-1,4,7-triyl)triacetic acid (NOTA). One example of this is the granzyme B imaging agent [^18^F]AlF-mNOTA-GZP developed by Goggi et al., which was easily synthesized via heating up the fluoride aluminium chloride with a chelator-modified tracer in a suitable buffer at 100 °C for 15 min [119]. One caveat is that the elevated temperature will only permit small molecule or peptide-like compounds to be directly labelled by this method. Pre-clinical data have shown that both [^68^Ga]Ga-mNOTA-GZP and [^18^F]AlF-mNOTA-GZP exhibit high specificity for granzyme B [119,146]. Tumour accumulation correlated with granzyme B expression in syngeneic mice treated with cancer immunotherapy.

Another PET tracer, [^64^Cu]Cu-DOTA-GRIP B, takes a very different approach for imaging granzyme B [123]. Instead of inhibiting the enzyme, Zhao et al. joined a membrane binding peptide and a masking peptide with the granzyme B cleavable sequence of IEPDVSQV. The cleavable link has the same IEPD sequence reported by Thornberry et al., which explains its high efficiency to cleavage. At its native state, the membrane accumulation sequence is protected by the masking peptide and does not have any membrane binding ability. Once the IEPDVSQV linker has been digested by granzyme B, the membrane binding peptide is activated and binds to any nearby cell membranes to affect tracing of granzyme B. Radiolabelling is carried out at a slightly elevated temperature of 50 °C for 30 min in a pH 7.0 buffer. The long half-life of ^64^Cu enables imaging at a later time point compared to ^18^F and ^68^Ga probes.

### 5.4. IFN-γ

Interferon gamma (IFN-γ) is a cytokine that is secreted by activated lymphocytes. Furthermore, IFN-γ plays a critical role in the activation of various immune cells and in the induction of anti-tumour immune response. Secreted IFN-γ promotes the polarization of macrophages towards a more pro-inflammatory and tumouricidal phenotype. In addition, IFN-γ results in the differentiation of T cells towards the Th1 subset, as well as the maturation of naïve T cells to effector CD8^+^ T cells [149]. Although IFN-γ was initially shown to inhibit B cell responses [150], its inhibitory effect is observed in pre-activated B cells and not resting B cells. IFN-γ controls the production of immunoglobulin isotypes by B cells; it increases the production of IgG2 and IgG3 by activated B cells while inhibiting the production of IgG, IgM, and IgE [151,152]. Furthermore, IFN-γ signalling leads to tumour cell death through mechanisms, such as the upregulation of HLA/MHC complex and Fas/FASL pathway, thus, making IFN-γ an attractive target for PET imaging of effector lymphocytes [153]. [^89^Zr]Zr-DFO-anti-IFN-γ mAb PET probe was found to detect increased levels of IFN-γ in tumour-bearing BALB/c mice receiving HER2/neu DNA vaccination [154]. In a model of induced T cell exhaustion, [^89^Zr]Zr-DFO-anti-IFN-γ uptake was found to be similar to isotype control, demonstrating a lack of anti-tumour T cell activity. Due to the limitations of mAb as imaging probes, the same group studied the pharmacokinetics and specificity of four ^89^Zr-labelled anti-IFN-γ diabodies. Only one radiolabelled diabody demonstrated promising in vitro and in vivo properties [155]. Nevertheless, further studies are necessary to achieve optimal imaging performance.

### 5.5. PET Imaging of Metabolic Targets Associated with Activated Immune Cells

Imaging of metabolic pathways that are associated with the activation of immune cells is an alternative potential method that could be used to determine therapeutic outcomes. Thymidine kinase 1 (TK1), deoxycytidine kinase (dCK), and deoxyguanosine kinase (dGK) have been studied as potential targets for PET imaging of immune cell activation [156,157]. Arabinofuranosylguanine (AraG), a substrate of mitochondrial dGK, is upregulated in activated T cells. The [^18^F]F-AraG probe, developed by Namavari et al. for imaging T cell activation and proliferation in cancer, employs comparable radiochemistry to the synthesis of [^18^F]FDG [158] (Figure 8). It utilizes a nucleophilic fluorination followed by the cleavage of the protecting group.

The specificity of [^18^F]F-AraG for activated T cells and its feasibility in predicting responses to anti-PD-1 therapy was observed in murine tumour models. A biodistribution study of [^18^F]F-AraG in six healthy volunteers demonstrated its safety [159]. Currently, [^18^F]F-AraG is being evaluated in patients with advanced NSCLC [160,161].

**Table 4 pharmaceutics-14-02040-t004:** Overview of PET probes under evaluation for imaging treatment-induced immune response.

Target	Name	Format	Radioisotope	Active Clinical Trial	Trial Number	Highlights
**PD-1**	[^89^Zr]Zr-DFO-nivolumab	mAb	^89^Zr	Phase 1 NSCLC	EudraCT: 2015-004760-11	Safe Uptake correlated with PD-1 expression
				Phase 2 Melanoma	NCT05289193	Recruiting
	[^89^Zr]Zr-DFO-pembrolizumab	mAb	^89^Zr	Phase 2 NSCLC	NCT03065764	Safe Tumour uptake was higher in patients that exhibited response to pembrolizumab treatment (not statistically significant) Tumour uptake did not correlate with PD-1 expression determined by IHC [162].
**PD-L1**	[^89^Zr]Zr-DFO-durvalumab	mAb	^89^Zr	Phase 2 HNSCC	NCT03829007	Safe and feasible Tracer did not predict treatment-induced immune response Tracer uptake did not correlate to PD-L1 expression [163].
				Phase 2 NSCLC	NCT03853187	Recruiting
	[^89^Zr]Zr-DFO-REGN3504	mAb	^89^Zr	Phase 1 Patients with advanced PD-L1 positive malignancies	NCT03746704	No published clinical data
	[^89^Zr]Zr-DFO-avelumab	mAb	^89^Zr	Phase 1 NSCLC	NCT03514719	No published clinical data
	[^68^Ga]Ga-NOTA-(hPD-L1)	nanobody	^68^Ga	Pre-clinical	-	Site-specifically radiolabelled High tumour uptake in PD-L1 expressing tumours Imaging is possible as early as 1-h post-injection [111].
	[^68^Ga]Ga-NOTA-Nb109	nanobody	^68^Ga	Pre-clinical	-	Non-blocking PET tracer Specifically binds to PD-L1 in various tumours High tumour uptake is observed at 10 min post-injection due to its small size [112,113].
	[^89^Zr]Zr-DFO-6E11	mAb	^89^Zr	Pre-clinical	-	Site-specific conjugation of DFO with glycan conjugation chemistry Detected PD-L1 expression in murine tumour models [105].
	[^68^Ga]Ga-THP-APN09 PET	nanobody	^68^Ga	Phase 1 Lung cancer, melanoma, and other solid tumours	NCT05156515	No published clinical data
**CTLA-4**	[^89^Zr]Zr-DFO-ipilimumab	mAb	^89^Zr	Phase 2 Metastatic melanoma	NCT03313323	Recruiting
**LAG-3**	[^89^Zr]Zr-DFO-REGN3767	mAb	^89^Zr	Phase 1 Relapsed/Refractory DLBCL	NCT04566978	Recruiting
**TIGIT**	[^68^Ga]Ga-NOTA-GP12	Peptide antagonist	^68^Ga	Pre-clinical	-	Possess high specificity and affinity for TIGIT Demonstrated high tumour uptake xenograft models [117].
				Tested in two patients with advanced NSCLC		No adverse effects were observed Moderate accumulation in tumour was observed Rapid clearance from circulation
**CD3**	[^89^Zr]Zr-DFO-anti-CD3	mAb	^89^Zr	Pre-clinical	-	High tumour uptake correlated with response to CTLA-4 immunotherapy in xenograft mouse model [129].
**CD4**	[^64^Cu]Cu-NOTA-IAB41	Minibody	^64^Cu	Pre-clinical	-	Specifically detects human CD4^+^ T cells without impacting their abundance, proliferation, and activation Can visualize various peripheral tissues in addition to orthotopically implanted GBM tumours [164].
	[^89^Zr]Zr-malDFO-GK1.5 cDb	Cys-Diabody	^89^Zr	Pre-clinical	-	Low-dose GK1.5 cDb yields high-contrast immune-PET images with minimal effects on T cell biology in vitro and in vivo and may be a useful tool for investigating CD4^+^ T cells in the context of preclinical disease models [165].
	^89^Zr-labelled anti-CD4 scFv	ScFv	^89^Zr	Pre-clinical	-	Can monitor the in vivo distribution of CD4^+^ T cells by immuno-PET [166].
**CD8**	[^89^Zr]Zr-DFO-IAB22M2C	Minibody	^89^Zr	Phase 1 Melanoma, lung, and hepatocellular carcinoma	NCT03107663	No side effects were observed High uptake was observed in spleen followed by bone marrow (CD8^+^ T-cell rich tissues) Uptake in tumour was detected at 2 h post-injection (most positive lesions were detectable by 24 h) [133].
				Phase 2 Advanced and metastatic solid malignancies	NCT03802123	Active (no published clinical data)
	[^68^Ga]Ga-NODAGA-SNA006a	Nanobody	^68^Ga	Phase 1		No adverse events Highest uptake was observed in spleen Uptake correlated with CD8 expression as confirmed by IHC [118].
				Phase 2		Recruiting
**OX40**	[^64^Cu]Cu-DOTA-AbOX40	mAb	^64^Cu	Pre-clinical	-	Both PET probes were demonstrated to detect treatment-induced immune response in murine models [136,137].
	[^89^Zr]Zr-DFO-OX40	mAb	^89^Zr			
**IL-2R**	[^18^F]FB-IL2	Small protein (cytokine)	^18^F	Phase 1	NCT02922283	Safe and feasible Did not detect treatment-related immune response [145].
	[^18^F]FBox-TTCO-IL2			Pre-clinical	-	No in vivo data published [141].
**Granzyme B**	[^18^F]AlF-mNOTA-GZP	Peptide	^18^F	Pre-clinical	-	Tumour uptake correlated with immune response in syngeneic tumour models Data demonstrated that pre-existing phenotypic abnormalities impact tracer uptake [119,146,147].
	[^68^Ga]Ga-mNOTA-GZP	Peptide	^68^Ga	Pre-clinical		
	[^64^Cu]Cu-DOTA-GRIP B	Peptide	^64^Cu	Pre-clinical		Tumour uptake correlated with tumoural granzyme B expression in syngeneic mouse model [123].
**IFN-γ**	[^89^Zr]Zr-DFO-anti-IFN-γ	mAb	^89^Zr	Pre-clinical	-	Tracer demonstrated specificity for IFN-γ Detects active anti-tumour immunity in situ in syngeneic murine models [154].
	[^89^Zr]Zr-DFO-NCS-anti-IFN-γ HL-11	Diabody	^89^Zr	Pre-clinical	-	Promising physiochemical properties were determined High tumour uptake was observed in syngeneic mouse model [155].
**AraG**	[^18^F]F-AraG	Small molecule (Nucleoside analog)	^18^F	Early phase 1 In healthy volunteers and patients with advanced NSCLC	NCT04678440	Recruiting
				Phase 1 cancer patients undergoing immunotherapy and/or radiation therapy	NCT03142204	

## 6. Conclusions

PET imaging of cytotoxic T cells is a powerful non-invasive method for characterising immune responses to cancer immunotherapies and, consequently, to aid clinical decision-making for cancer treatment. Despite the growing success of many immunotherapeutic agents, they still face challenges. Therefore, the identification of novel predictive biomarkers and the characterisation of the TME may improve patient selection and treatment evaluation. In this review, we discussed the most recent developments in PET imaging of immune response. The high sensitivity of PET in combination with T cell-specific probes enables quantification of cytotoxic T cell dynamics. Although PET tracers for CD8^+^ and CD3^+^ capture T cell dynamics, they do not provide information on the functional state of cytotoxic T cells. Therefore, tracers targeting IL-2R, granzyme B, OX40, IFN-γ, and AraG may provide more comprehensive information on treatment responses. However, some PET imaging approaches have been associated with unclear results as seen for IL-2R, consequently hindering the clinical application of these tracers. Nevertheless, multiple PET tracers targeting activation/exhaustion markers are being developed and evaluated in clinical studies.

## Figures and Tables

**Figure 1 pharmaceutics-14-02040-f001:**
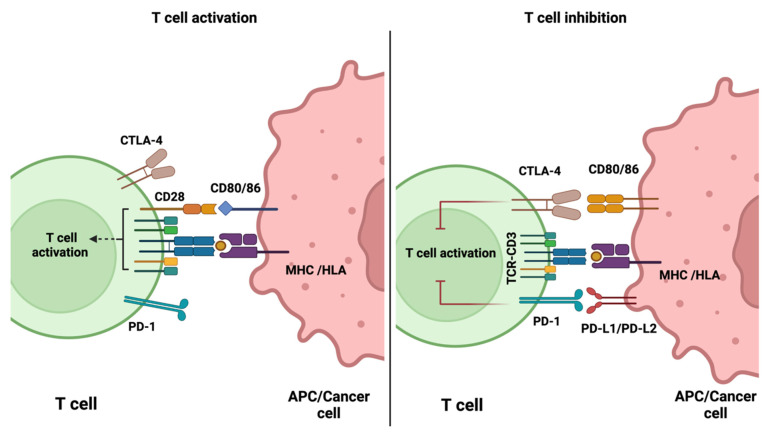
Schematic diagram demonstrating the role of immune checkpoint molecules in the regulation of activated T cells. Diagram was generated using BioRender.com (accessed on 17 August 2022).

**Figure 2 pharmaceutics-14-02040-f002:**
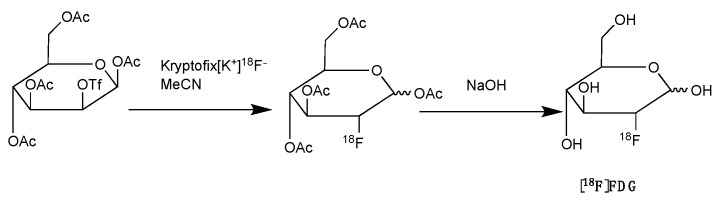
Synthesis of [^18^F]FDG with nucleophilic fluorination followed by strong base deprotection.

**Figure 3 pharmaceutics-14-02040-f003:**
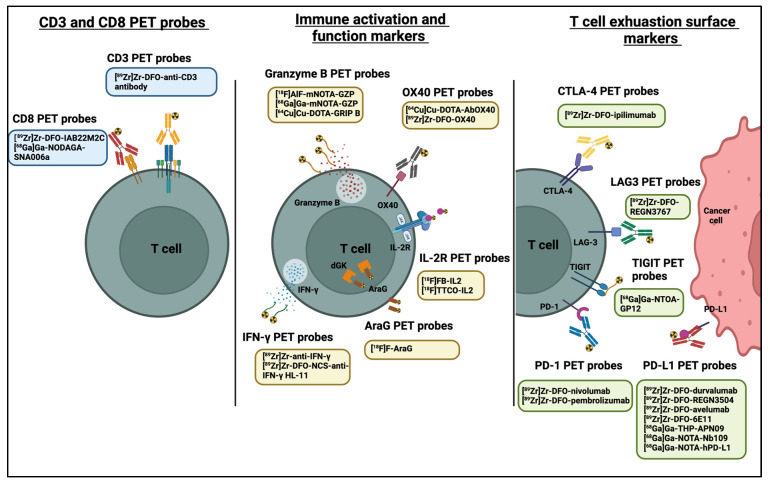
PET probes used in predicting and evaluating immune response to anti-cancer immunotherapy. Diagram was created with BioRender.com (accessed on 28 July 2022).

**Figure 4 pharmaceutics-14-02040-f004:**
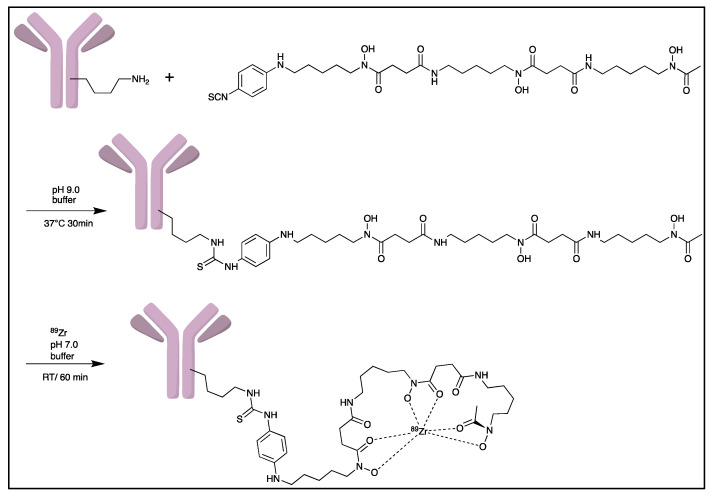
General procedure for p-NCS-Bz-DFO conjugation and ^89^Zr radiolabelling with antibody.

**Figure 5 pharmaceutics-14-02040-f005:**
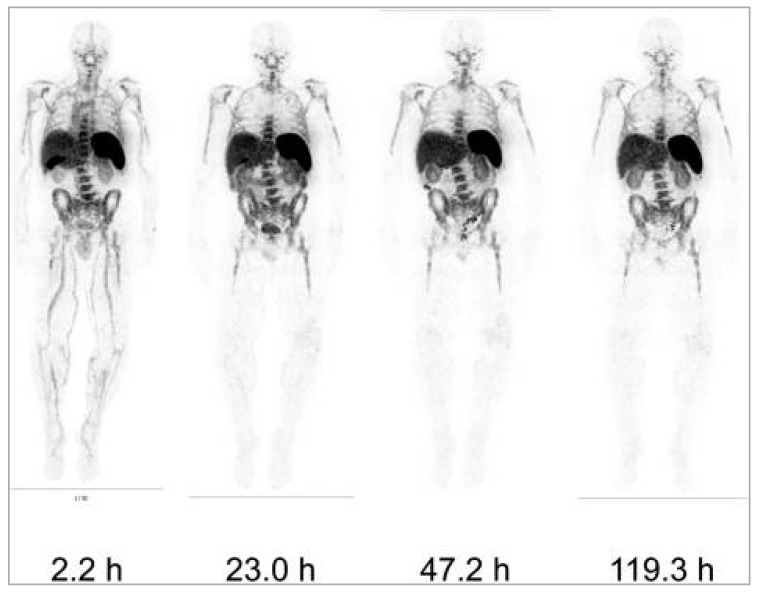
Whole-body images of a 1.5 mg minibody dose of [ ^89^Zr]Zr-DFO-IAB22M2C at different time points post-injection in one patient. Reproduced from Pandit-Taskar et al. [133].

**Figure 6 pharmaceutics-14-02040-f006:**
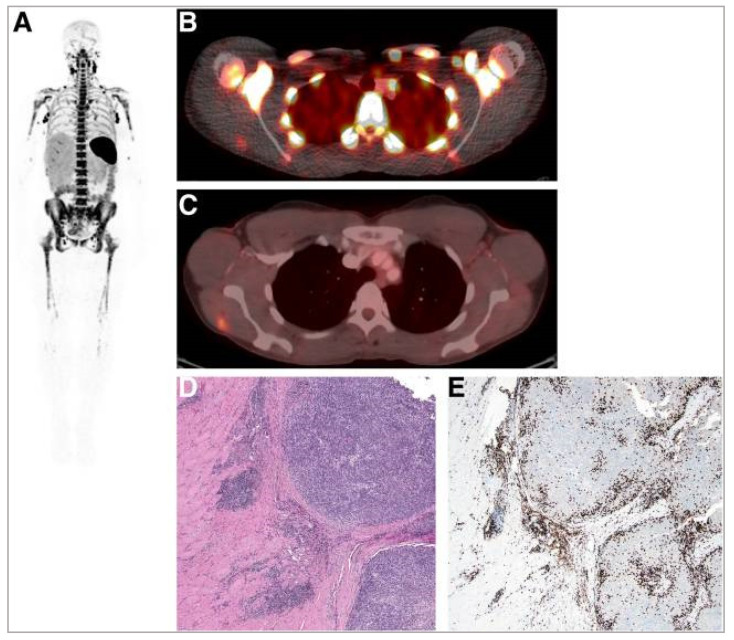
A whole-body image (maximum-intensity projection) in one patient at 24 h post injection of 0.2 mg of [^89^Zr]Zr-DFO-IAB22M2C demonstrating intense signals in lymph nodes (**A**). Fusion image demonstrates the uptake of [^89^Zr]Zr-DFO-IAB22M2C in lesions in the deltoid (**B**) that also showed [^18^F]FDG uptake (**C**). (**D**) Haematoxylin- and eosin-stained section demonstrates melanoma tumour nodules on right within skeletal muscle. (**E**) Immunohistochemistry staining shows the presence of CD8^+^ T cells at the periphery and infiltrating tumour. Reproduced from Pandit-Taskar et al. [133].

**Figure 7 pharmaceutics-14-02040-f007:**
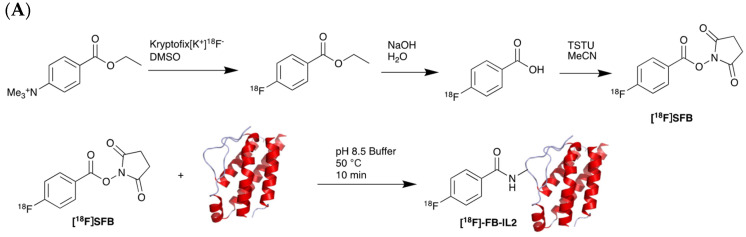
Procedure for radiolabelling of interleukin-2 (IL-2). (**A**) [^18^F]SFB radiolabelling of IL-2 starting with nucleophilic fluorination of 4-(Ethoxycarbonyl)-N,N,N-trimethylbenzenaminium. (**B**) [^18^F]Box-Tz radiolabelling of IL-2 enabled by the click chemistry between TCO labelled IL-2 tetrazine functionalised [^18^F]FBox-Tz.

**Figure 8 pharmaceutics-14-02040-f008:**
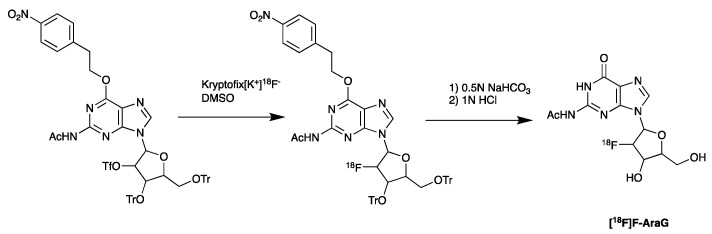
Synthesis of [^18^F]AraG, starting with nucleophilic fluorination of the protected precursor, followed by acid deprotection.

**Table 1 pharmaceutics-14-02040-t001:** Common radionuclides and their physical properties.

Radionuclide	Half-Life	Decay Mode (β^+^ Mode %)	Position Energy (MeV)	Production Method
^11^C	20.4 min	99	0.97	^14^N(p, α)^11^C
^18^F	109.7 min	97	0.65	^18^O(p, n)^18^F
^68^Ga	67.7 min	89	1.9	^68^Ge/^68^Ga(generator)
^44^Sc	3.97 h	94	1.47	^44^Ca(p, n)^44^Sc or ^44^Ti/^44^Sc (generator)
^64^Cu	12.7 h	18	0.65	^64^Ni(p, n)^64^Cu
^89^Zr	78.4 h	23	0.91	^89^Y(p, n)^89^Zr
^124^I	100.2 h	23	1.54	^124^Te(p, n)^124^I

**Table 2 pharmaceutics-14-02040-t002:** Radionuclides and their corresponding chelators.

	Chelator	Complex	Labelling Conditions
^68^Ga			
NOTA	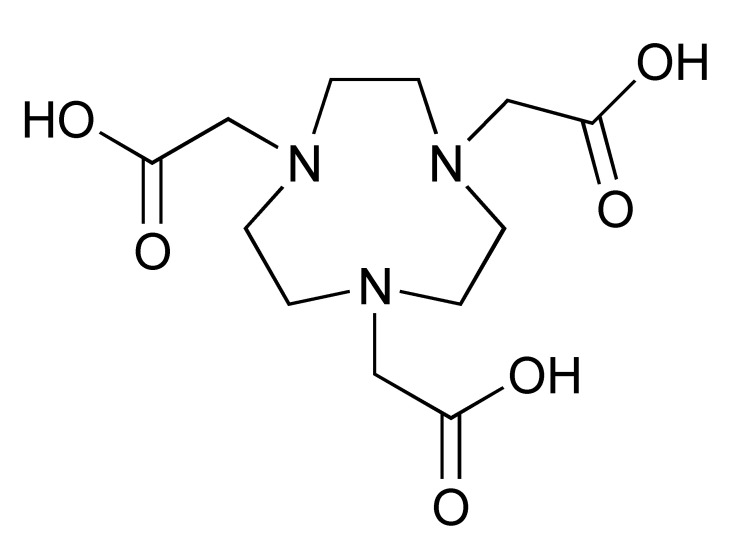	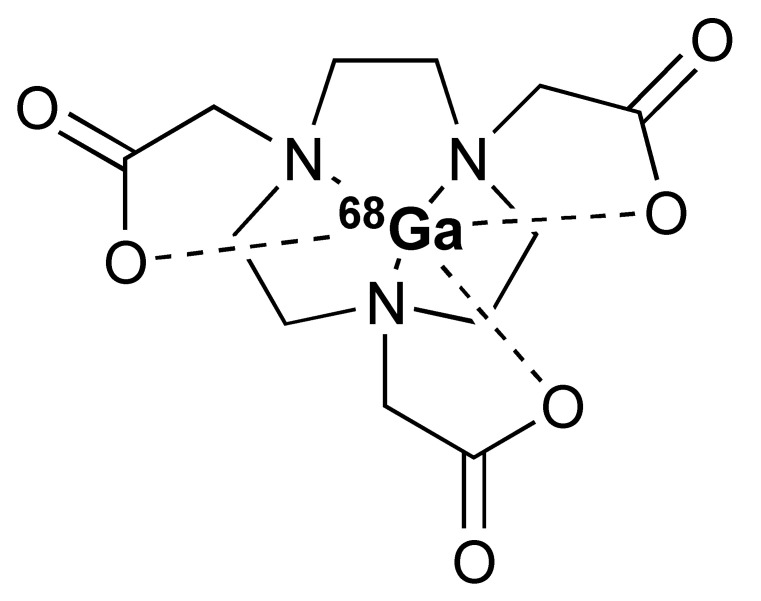	RT/pH 4.0/30 min
DOTA	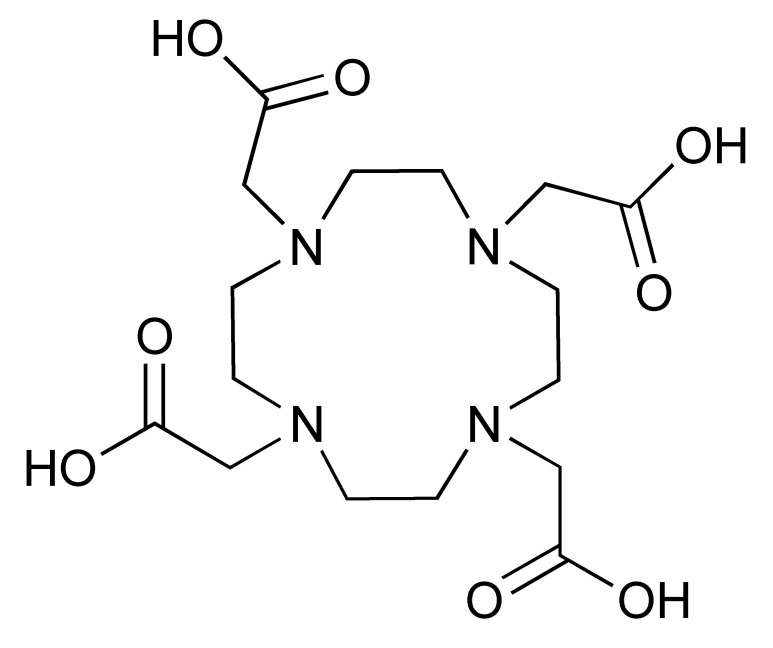	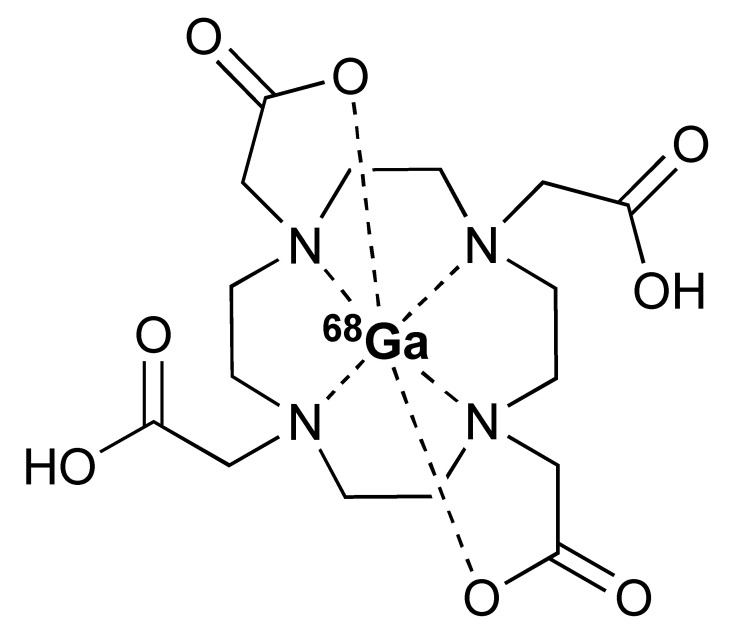	95 °C/pH 4.0/6.6 min
THP	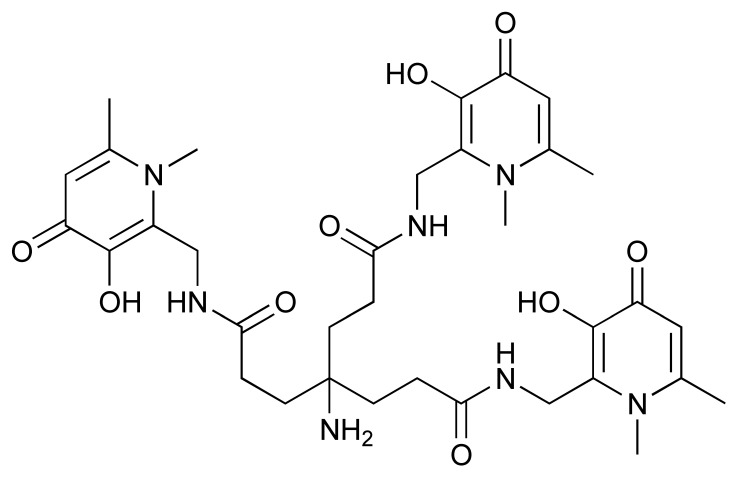	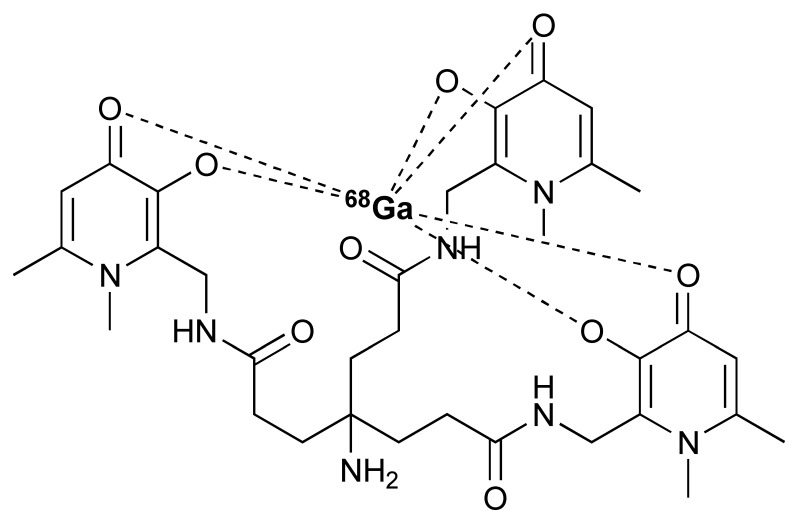	RT
^64^Cu			
NOTA	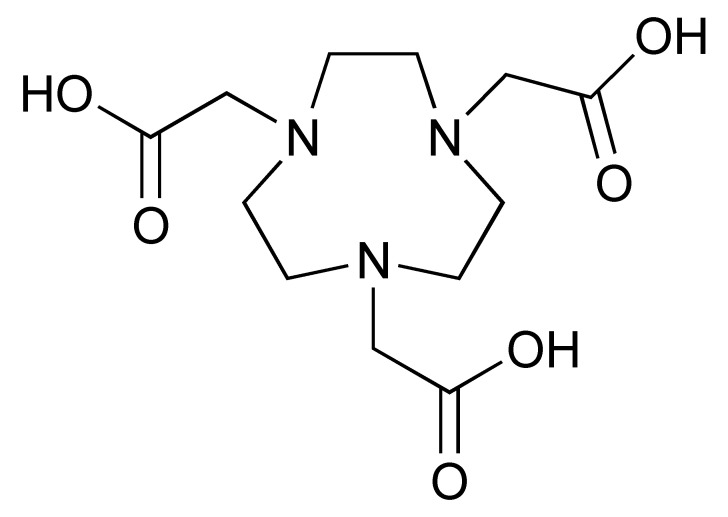	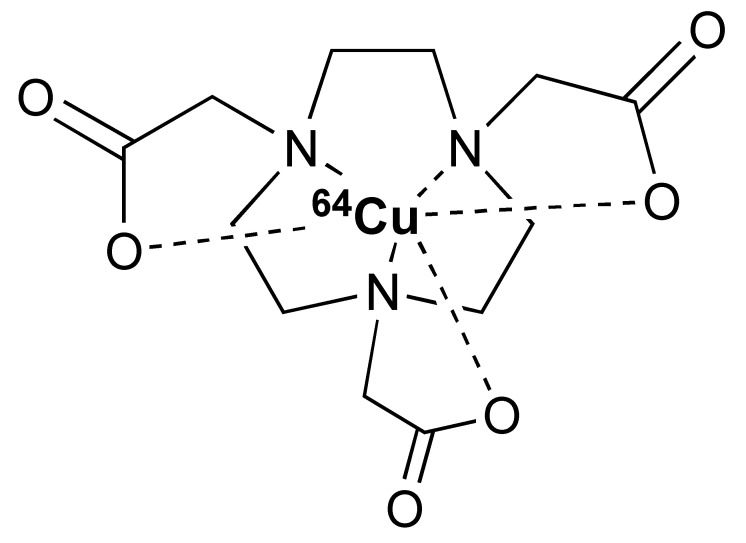	40 °C/pH 6.5/30 min
DOTA	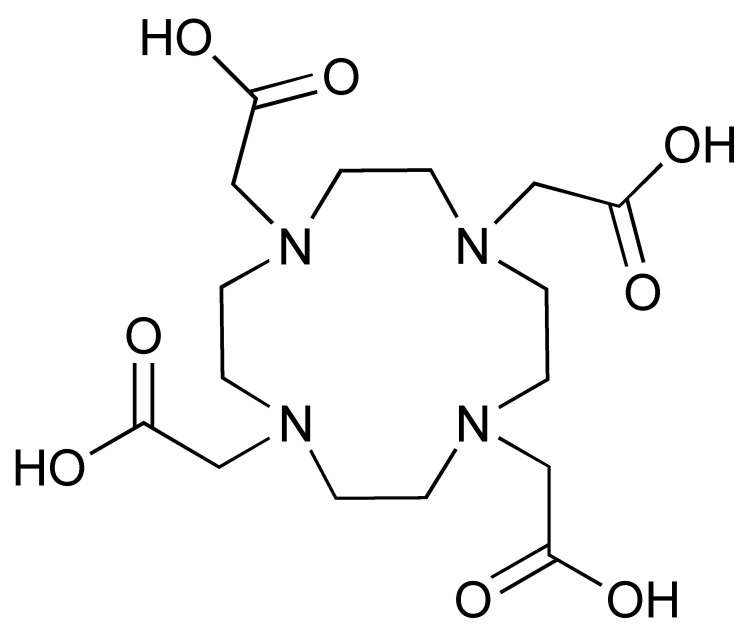	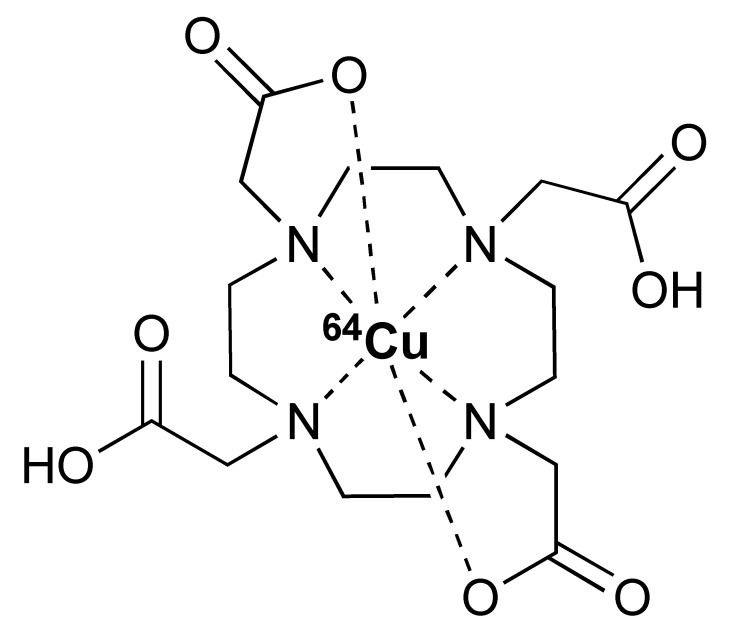	37 °C/pH 5.5/60 min or 50 °C/pH 7.0/30 min
^89^Zr			
DFO	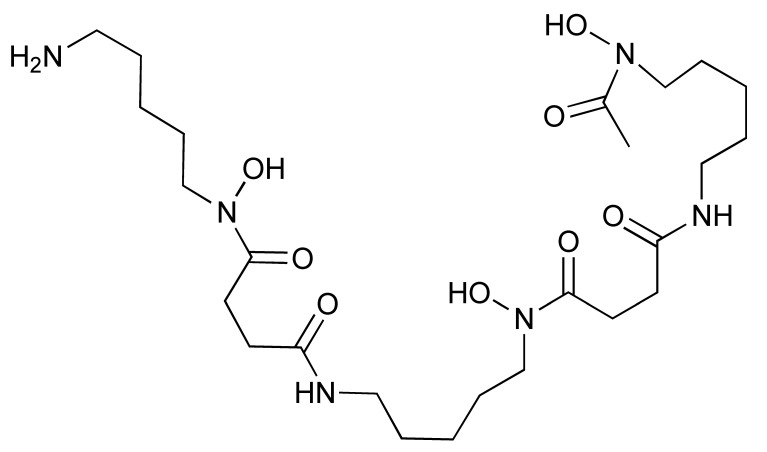	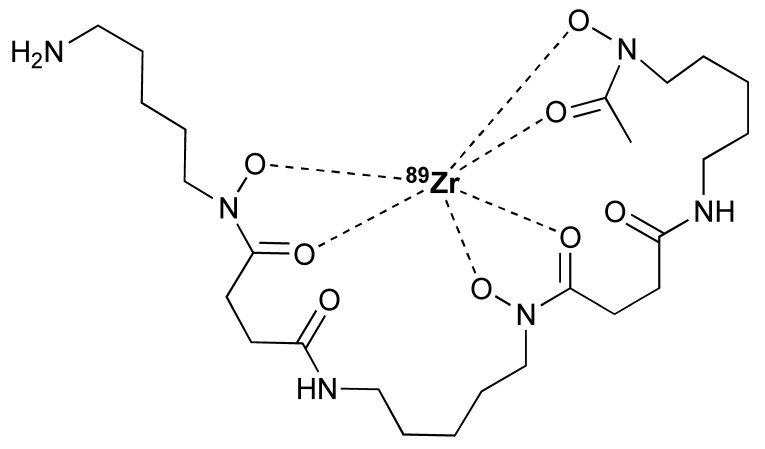	RT/pH 6.8–7.5/60 min

**Table 3 pharmaceutics-14-02040-t003:** Examples of PET tracers for cytotoxic T-cell and their highlights.

Targeting Motif	Probes	Highlights
Small molecule (MW < 1000 Da)	[^18^F]F-AraG	Very fast pharmacokinetics Tolerate very harsh radiosynthesis condition
Peptide (MW 1000~3000 Da)	[^68^Ga]Ga-mNOTA-GZP [^18^F]AlF-mNOTA-GZP [^68^Ga]Ga-NOTA-GP12 [^64^Cu]Cu-DOTA-GRIP B	Very fast pharmacokinetics Ease of chemical synthesis Tolerate harsh radiosynthesis condition
Nanobody (MW ~15 kDa)	[^68^Ga]Ga-NOTA-(hPD-L1) [^68^Ga]Ga-NOTA-Nb109 [^68^Ga]Ga-NODAGA-SNA006a	Fast pharmacokinetics Increasingly easy for manufacture/obtain Moderately tolerate to harsh radiosynthesis conditions High binding affinity
Diabody (MW~55 kDa)	[^89^Zr]Zr-DFO-NCS-anti-IFN-γ HL-11	Intermediate pharmacokinetics High binding affinity
Minibody (MW~80 kDa)	[^89^Zr]Zr-Df-IAB22M2C	Intermediate pharmacokinetics Higher tumour uptake compared to fragments High binding affinity Prefer liver as metabolic organ
mAb (MW~150 kDa)	[^89^Zr]Zr-DFO-nivolumab [^89^Zr]Zr-DFO-pembrolizumab [^89^Zr]Zr-DFO-durvalumab [^89^Zr]Zr-DFO-REGN3504 [^89^Zr]Zr-DFO-avelumab [^89^Zr]Zr-DFO-6E11 [^89^Zr]Zr-DFO-ipilimumab [^89^Zr]Zr-DFO-REGN3767 [^64^Cu]Cu-DOTA-AbOX40 [^89^Zr]Zr-DFO-OX40 [^89^Zr]Zr-DFO-anti-IFN-γ	Slow pharmacokinetics, long circulation half-life Relatively easy for manufacture/obtain High tumour uptake High binding affinity Prefer liver as metabolic organ

## Data Availability

Not applicable.

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
