# Peer review of "Positron Emission Tomography Probes for Imaging Cytotoxic Immune Cells"

_pharmaceutics, 2022, doi:10.3390/pharmaceutics14102040_

Round 1
Reviewer 1 Report
The work carried out by the authors in this review about PET imaging for immune response is quite interesting as it gives a comprehensive overview on the specific use of this technique to identify the best biomarkers and therefore have a better prevention and treatment in cancer immunotherapy. There are just a few minor comments to consider:
- The abstract is an important part of the review since it is the first door for other researchers to be interested in reading it. I consider that it is too brief and general, perhaps it would be better to specify some aspects that are explained in detail in this review.
- Each of the parts presented are quite clear and well explained, except for the therapeutic vaccines section which contrasts with what was written before and after this. Perhaps it would fit better to talk about it within the challenges associated with cancer immunotherapy.
- In the section of OX40, the next review “doi: 10.1111/j.1600-065X.2009.00766.x” have to be cited, which talk about the significant of this TNF receptor in immune disease.
- The section on IFN-x can be extended a little further. Its role in the induction of IgG2 and IgG3 production by activation of plasma B cells should be discussed.
- Line 304, there is an error referencing Table1 or Table 2.
- Line 342, a comma is missing, and the capital letter should be removed.
- Line 408, the dot should be after quoting Figure 4.
- In Figure 4, the degree symbol has not been indicated in the first part of the reaction.
Reviewer 2 Report
The manuscript submitted by Amgheib A., et al. is a review article on the development of radiolabeled tracers for imaging cytotoxic immune cells with positron emission tomography (PET). Immunotherapy is emerging as an effective cancer treatment option but only a small group of patients respond to the treatment. Non-invasive imaging technology such as PET has the capability to identify patients who are more likely to respond to the treatment, and can greatly facilitate the application of immunotherapy in the clinic. This manuscript provides a very comprehensive review on the development of PET tracers for imaging cytotoxic immune cells with the aim to select patients for treatment. This review article is well written, and the publication of this article in the great interest of Pharmaceutics readers who are in the field of radiopharmaceutical sciences. Listed below are some minor suggested changes:
· Line 304: delete “Table 2”.
· Table 1: Change “b+ mode” to “β+ mode”.
· Table 2: The chemical structures of Ga-DOTA, Cu-NOTA and Cu-DOTA are not accurate. Please see the correct structures as published by Wadas TJ, et al. in Chem Rev 2010; 110: 2858-2902.
· Table 3: the “circulation” in the right lower cell is written in a different font.
· Line 336: “As a forementioned” should be “As aforementioned”.
· Line 415: “anti-PDL1” should be “anti-PD-L1”.
· Fig 4: the chemical structures of all DFO-Bz-NCS are not accurate (see https://www.chematech-mdt.com/produit/p-ncs-bz-dfo/ ).
· Line 542: the “-8” in “Kd~10-8 M” should be superscript.
· References # 158-161 (Lines 913-919) are written in a different font.
